# Dendritic Growth Optimization: A Novel Nature-Inspired Algorithm for Real-World Optimization Problems

**DOI:** 10.3390/biomimetics9030130

**Published:** 2024-02-21

**Authors:** Ishaani Priyadarshini

**Affiliations:** School of Information, University of California, Berkeley, CA 94720, USA; ishaani@berkeley.edu

**Keywords:** optimization, nature-inspired, DGO, machine learning, generalizability

## Abstract

In numerous scientific disciplines and practical applications, addressing optimization challenges is a common imperative. Nature-inspired optimization algorithms represent a highly valuable and pragmatic approach to tackling these complexities. This paper introduces Dendritic Growth Optimization (DGO), a novel algorithm inspired by natural branching patterns. DGO offers a novel solution for intricate optimization problems and demonstrates its efficiency in exploring diverse solution spaces. The algorithm has been extensively tested with a suite of machine learning algorithms, deep learning algorithms, and metaheuristic algorithms, and the results, both before and after optimization, unequivocally support the proposed algorithm’s feasibility, effectiveness, and generalizability. Through empirical validation using established datasets like diabetes and breast cancer, the algorithm consistently enhances model performance across various domains. Beyond its working and experimental analysis, DGO’s wide-ranging applications in machine learning, logistics, and engineering for solving real-world problems have been highlighted. The study also considers the challenges and practical implications of implementing DGO in multiple scenarios. As optimization remains crucial in research and industry, DGO emerges as a promising avenue for innovation and problem solving.

## 1. Introduction

In the dynamic evolution of machine learning and data science, optimization is indispensable. Whether fine-tuning the hyperparameters of a deep neural network, optimizing supply chain logistics, or finding the most efficient route for delivery services, the need for efficient optimization techniques is abundant. To meet this demand, researchers and practitioners have turned to nature-inspired optimization methods, a class of algorithms that draw inspiration from natural processes to tackle complex optimization problems [1,2,3,4,5]. The diversity and complexity of optimization problems in scientific research and practical applications necessitate innovative solutions. While effective in many scenarios, traditional optimization methods often face challenges in handling high-dimensional, multi-modal, and noisy data. These limitations have led to the need for alternative techniques, and nature-inspired optimization has emerged as a promising paradigm. Historically, optimization has been approached using mathematical techniques like gradient descent, simulated annealing, and linear programming, each with advantages and limitations [6,7,8,9,10]. Gradient descent, for instance, is a powerful method for convex optimization problems but can get trapped in local optima in non-convex landscapes. Simulated annealing introduces randomness to escape local optima but may require extensive computational resources. Linear programming excels in linear optimization problems but is less suited for nonlinear scenarios. These traditional methods often struggle with high-dimensional optimization, where the search space is vast, and evaluating potential solutions is computationally expensive. Moreover, problems in real-world applications are rarely linear or noise-free, making it challenging for these classical techniques to deliver optimal solutions consistently. Nature-inspired optimization methods have gained prominence due to their ability to address these limitations [11,12,13]. Algorithms like Genetic Algorithms, Particle Swarm Optimization, Ant Colony Optimization, and Simulated Annealing draw inspiration from biological, ecological, and physical processes. For example, Genetic Algorithms mimic natural selection and genetic variation to evolve solutions. Particle Swarm Optimization models the collective behavior of birds or fish to find optimal solutions collectively [14]. The foraging behavior of ants inspires Ant Colony Optimization to discover efficient paths. These methods have been successfully applied in diverse fields. Genetic Algorithms have been used for feature selection In machine learning, antenna design, and scheduling problems. Particle Swarm Optimization has found applications in optimizing neural network parameters, clustering, and robotics. Ant Colony Optimization has been applied to network routing, vehicle routing, and data clustering [15,16]. However, despite their successes, these nature-inspired optimization methods are not without their limitations. They may struggle with problems involving high-dimensional solution spaces, and their convergence may be slow in complex landscapes [17,18,19]. This is where Dendritic Growth Optimization (DGO) steps in, offering a novel optimization approach that overcomes these challenges. DGO is a nature-inspired optimization algorithm inspired by the branching patterns observed in natural systems. It represents a paradigm shift in optimization by harnessing the principles of dendritic growth. In nature, dendrites are branching structures found in various contexts, from the growth of trees to the formation of snowflakes. DGO capitalizes on dendritic structures’ branching and connection mechanisms to explore solution spaces and facilitate information sharing among potential solutions.

DGO distinguishes itself by offering a novel strategy for optimization. Instead of relying solely on iterative updates or evolutionary processes, DGO uses a branching and connection approach. This strategy enables DGO to efficiently navigate solution spaces by exploring multiple paths simultaneously. Moreover, DGO fosters collaboration among potential solutions, allowing them to exchange information and collectively improve. This study delves into the concept, architectural work, and applications of DGO. The primary objectives of this study include introducing DGO, explaining its unique approach, and demonstrating its capabilities through empirically validating existing datasets. DGO’s foundation lies in the principles of dendritic growth, a phenomenon observed in countless natural systems, and sets it apart. DGO leverages the wisdom of nature to navigate complex landscapes and find optimal solutions efficiently. What sets DGO apart and positions it as a promising optimization technique is its adaptability to high-dimensional, multi-modal, and noisy landscapes. DGO’s branching and connection strategy allows it to explore diverse solution spaces effectively. Moreover, its collaborative nature enables it to overcome challenges in finding global optima in complex, non-convex landscapes. The main contributions of this paper are as follows:DGO is a novel addition to the optimization toolkit. It diverges from traditional methods and existing nature-inspired algorithms by capitalizing on dendritic growth principles.The architecture of DGO is inherently distinctive. Inspired by the branching and connection mechanisms observed in dendritic structures, it introduces an innovative way of navigating complex solution spaces.The study details how the proposed strategy solves the local minima problem and finds a global optimum.In this paper, DGO is not limited to theoretical concepts but is applied practically to various datasets across different domains. This pragmatic approach demonstrates adaptability and relevance in diverse problem-solving scenarios and validates its generalizable nature.DGO’s performance is rigorously assessed using a variety of evaluation metrics. The study does not limit ourselves to a single criterion but provide a comprehensive analysis, ensuring a thorough understanding of its strengths and limitations.A list of the study’s applications, potential challenges, and limitations has been presented. By openly discussing these aspects, the study provides a balanced view of its capabilities and areas for potential improvement.

The rest of the paper is organized as follows. The study discusses the related research works and methodology in Section 2. The methodology section primarily highlights the working of DGO its and its architecture. Section 3 of the results discusses the datasets considered for the study, followed by a performance evaluation of ML algorithms with and without DGO. The study also explores other optimization techniques to understand the strengths and weaknesses of DGO. A comparative analysis of this work with some previous works follows this. Finally, a conclusion is added to the paper.

## 2. Materials and Methods

In this section, some of the related works from past research have been presented on nature-inspired optimization algorithms. This section also summarizes the limitations of these existing methods to validate the robustness of the proposed technique. This is followed by the methodology section, where the architecture and working of the optimization model are proposed, with the steps, mathematical representation, and pseudocode. This section also discusses how DGO can solve the local minima problem and find a global optimum.

### 2.1. Related Works

Mahadeva et al., 2023 [20] proposed an ANN-based modified whale optimization strategy for reverse-osmosis desalination plants. The proposed technique uses literature-derived datasets with input parameters to predict permeate flux (0.118–2.656 L/h m²). The specific features of the dataset include flow rate, temperature, concentration, etc. Ten MWOA-ANN models outperform existing approaches, minimizing errors and overfitting risks. Model-6, featuring a single hidden layer with eleven nodes, excels with a 99.1% regression coefficient (R^2^) and minimal errors (MSE = 0.005), making it a promising tool for industrial plant designers. Some limitations of the study may be in the form of the dataset since the study relies on datasets collected from the literature, which may introduce biases or variations in data quality. Moreover, the effectiveness of the MWOA-ANN models may be context-specific to the RO desalination plant considered in this study. Hence, although the hybridized model shows promise, several concerns exist regarding the overall training time and resource consumption. Zhao et al., 2023 [21] introduced the Sea-Horse Optimizer (SHO), a novel swarm intelligence metaheuristic inspired by sea horses’ natural behaviors. SHO comprises three stages mimicking sea horse movement, predation, and unique reproduction traits. It strikes a balance between local and global exploration. Evaluation of various functions and benchmarks and real-world engineering problems confirms SHO’s high performance and adaptability, making it a promising optimizer for diverse applications. Although SHO shows promise in simulation, its practical implementation in real-world scenarios may involve additional challenges and uncertainties. The study’s effectiveness in handling constraint problems is mentioned but not extensively explored, warranting further investigation. Sahoo et al., 2023 [22] introduced the modified Dynamic Opposite Learning-based MFO (m-DMFO) algorithm, incorporating a Dynamic Opposite Learning (DOL) strategy to address issues like premature convergence and local minima trapping. The m-DMFO algorithm is rigorously assessed on various benchmark functions, including IEEE CEC’2014 tests, and compared with multiple optimization algorithms. Statistical tests, convergence analysis, and diversity measurements confirm the m-DMFO’s robustness and superior performance, with results exceeding 90% success rates. The algorithm also excels in Friedman and Wilcoxon rank tests and demonstrates its effectiveness in solving real-world engineering problems, reaffirming its enhanced performance. While the m-DMFO algorithm outperforms other methods in many instances, the statistical significance of these differences may not be thoroughly explored. Dhal et al., 2023 [23] suggested a novel optimization technique to address noise susceptibility and high computation time in Fuzzy C-means (FCM) image segmentation. It introduces the histogram-based fast fuzzy image clustering (HBFFIC) method, utilizing morphological reconstruction (MR) for noise immunity and gray-level histogram clustering to reduce computational overhead. However, HBFFIC can still be prone to local optima due to FCM’s arbitrary initialization. To mitigate this, the study employs nature-inspired optimization algorithms (NIOAs), specifically the Archimedes optimizer (AO), to identify optimal cluster centers. Experimental results using real-world images, including synthetic, grayscale, and color pathology images, demonstrate that the proposed hybrid algorithm (HBFFIC-AO) surpasses state-of-the-art methods regarding segmentation accuracy, comparison score, etc., establishing its superiority in noisy image segmentation. While the proposed hybrid algorithm (HBFFIC-AO) aims to enhance noise immunity, the extent to which it can handle different types and levels of noise remains unexplored. Hu et al., 2023 [24] analyzed the Chameleon Swarm Algorithm (CSA) inspired by chameleon foraging strategies, demonstrating competitive performance with other algorithms. However, CSA faces challenges like limited exploitation ability, susceptibility to local optima, and low convergence accuracy in complex applications. The Modified Chameleon Swarm Algorithm (MCSA), incorporating fractional-order calculus, sinusoidal parameter adjustment, and a comprehensive learning strategy (CCL), was introduced to address these issues. Fractional-order calculus enhances local search, while sinusoidal parameter adjustment balances exploration–exploitation. The CCL strategy promotes diversity and avoids local optima. MCSA outperforms CSA and other algorithms on benchmark functions and engineering designs, showcasing its competitive capabilities for solving optimization problems. While the Modified Chameleon Swarm Algorithm (MCSA) introduces various enhancements, the sensitivity of these modifications to parameter settings, which could impact performance in different scenarios, is not extensively discussed. Seyyedabbasi and Kiani 2023 [25] introduced the Sand Cat Swarm Optimization (SCSO) algorithm, inspired by the sand cat’s survival behavior in nature, which includes detecting low frequencies and efficient prey hunting. SCSO comprises two key phases, balancing exploration and exploitation effectively. It excels in finding solutions with fewer parameters and operations through adaptive strategies, demonstrating its versatility. The algorithm is evaluated using 20 well-known and 10 complex CEC2019 benchmark functions, outperforming other metaheuristic algorithms by finding the best solutions in 63.3% of the test cases. Additionally, SCSO proves successful in seven challenging engineering design problems, displaying superior convergence rates and optimal solution identification compared to other methods. The evaluation primarily relies on benchmark test functions and engineering design problems, potentially overlooking the complexities of real-world scenarios. Mohammed and Rashid 2023 [26] proposed the Fox optimizer (FOX), a novel nature-inspired algorithm mimicking fox foraging behavior. FOX efficiently measures distances to execute jumps for prey capture. The algorithm is evaluated on classical benchmark functions and CEC2019 benchmarks, outperforming optimization strategies like Dragonfly, Particle Swarm, Grey Wolf, Whale Optimization, Generic Algorithms, etc. Statistical tests confirm FOX’s superiority. Parameter sensitivity analysis reveals its diverse exploratory and exploitative behaviors. FOX successfully tackles engineering problems like pressure vessel design and economic load dispatch, consistently outperforming the other algorithms considered for the study. The practical applicability of FOX to a broader range of real-world problems and industries requires further exploration. Yuan et al., 2023 [27] suggested the Coronavirus Mask Protection Algorithm (CMPA), inspired by COVID-19 prevention methods. CMPA mimics the infection and immunity phases, akin to mask-wearing and social distancing, crucial for human self-protection. It is mathematically modeled and evaluated against benchmark functions, CEC2020 problems, and truss design challenges, outperforming state-of-the-art optimizers statistically. Additionally, CMPA enhances the mass and deflection of a gantry crane’s main girder by 16.44% and 7.49%, respectively, showcasing its practical applicability. CMPA’s evaluation primarily focuses on its performance and may not be directly generalized to other optimization methods or complex real-world scenarios. Moreover, the algorithm’s effectiveness could be sensitive to parameter settings, requiring careful tuning for different problem domains. Agushaka et al., [28] proposed the novel Gazelle Optimization Algorithm (GOA), drawing inspiration from gazelles’ survival strategies against predators. GOA operates in two phases, mirroring gazelles’ behavior: an exploitation phase simulates grazing, while an exploration phase involves evading predators. These phases iteratively work towards finding optimal solutions to diverse optimization problems. Extensive testing on benchmark functions and engineering design challenges showcases GOA’s superiority over nine state-of-the-art algorithms, confirmed by statistical analysis. These results affirm GOA’s potential as a robust optimization tool adaptable to various domains. GOA’s performance may vary across optimization problems and require fine-tuning for optimal results. Abdel-Basset et al., 2023 [29] recommended the Nutcracker Optimization Algorithm (NOA), inspired by Clark’s nutcrackers, for emulating their distinct behaviors in two seasons, i.e., searching for seeds and spatial-memory-based cache retrieval. NOA employs diverse local and global search operators for robust optimization, demonstrating superior performance. It is evaluated on various benchmarks, including CEC-2014, CEC-2017, CEC-2020, and real-world engineering problems, outperforming recent, highly cited, and competition-winning algorithms like LSHADE-cnEpSin, LSHADE-SPACMA, AL-SHADE, and L-SHADE. NOA ranks first and achieves remarkable results, solidifying its status as an effective optimization tool. The study introduces NOA as inspired by the specific behaviors of nutcrackers. It does not explore potential variations or modifications of NOA to address different optimization scenarios or adapt to diverse problem characteristics. Hashim and Hussien 2023 [30] proposed a Snake Optimizer (SO), a novel metaheuristic algorithm inspired by snake mating behavior. SO is designed to handle various optimization tasks efficiently, mathematically modeling snake foraging and reproduction behaviors. Compared to nine other algorithms, it is evaluated on 29 CEC 2017 benchmark functions and real-world engineering problems. Experimental results demonstrate SO’s effectiveness in balancing exploration, exploitation, and rapid convergence across diverse landscapes. The limitations of this study include the need for further research to assess SO’s performance on more complex and large-scale optimization problems. Additionally, conducting experiments on a broader range of real-world engineering applications would provide a more comprehensive understanding of its practical utility. Zhong et al., 2022 [31] introduced the Beluga Whale Optimization (BWO) algorithm, inspired by beluga whale behaviors, to tackle optimization problems. BWO encompasses exploration, exploitation, and whale fall phases, akin to the whales’ pair swim, prey, and whale fall activities. Self-adaptive balance factors and probability control govern exploration and exploitation, while Levy flight enhances global convergence. BWO’s effectiveness is evaluated using 30 benchmark functions, showing competitiveness across unimodal and multi-modal optimization problems. It ranks first in scalability analysis among comparable metaheuristic algorithms via the Friedman ranking test. Additionally, BWO exhibits promise in solving complex real-world engineering problems. While BWO shows promise, further research is needed to assess its performance on a wider range of complex optimization problems and its applicability to various domains. The algorithm’s parameter settings may require tuning for optimal results in different applications, which could pose a challenge in practice.

A majority of previously proposed optimization algorithms exhibit limitations in areas such as training time, resource consumption, extensive exploration, varied parameter settings in different scenarios, and practical applicability to real-world problems. DGO effectively addresses and mitigates these issues, outperforming many ML algorithms, DL algorithms, and metaheuristic optimization algorithms.

### 2.2. Methodology

DGO working Optimization is a fundamental challenge across various domains, and Dendritic Growth Optimization (DGO) stands at the forefront of innovative techniques designed to address complex optimization problems. Drawing inspiration from the branching and connection mechanisms observed in dendritic growth processes in nature, DGO offers a fresh perspective on problem solving.

DGO is based on the principles of branching and connection. Dendritic growth is characterized by the iterative branching of structures, much like the branches of a tree extending in multiple directions. DGO mimics this phenomenon by branching and exploring multiple paths simultaneously. Let us dive into the mechanics of how DGO works.

Step 1: Initialization of seeds-DGO begins with initializing a set of seeds, each representing a potential solution. These seeds serve as starting points in the search for optimal solutions. In mathematical terms, the seeds can be represented as a set of vectors: {*S* = *S*_1_, *S*_2_, *S*_3_, *S*_4_, …, *S_n_*} Here, *n* represents the number of seeds, and each *S_i_*, is a vector representing a potential solution.

Step 2: Branching Strategy—DGO employs a branching strategy to explore the solution space more effectively. For each seed *S_i_*, the algorithm generates multiple branches, each representing a slight perturbation or variation in the original seed. The branching process introduces diversity into the search, enabling the exploration of different solution paths. Mathematically, the branching operation can be represented as follows: *B*(*S_i_*) = *{B_i_*_1_, *B_i_*_2_, *B_i_*_3_, *B_i_*_4_, …, *B_im_*}, where *B*(*S_i_*) is a set of branches generated from seed *S_i_*, and *m* is the number of branches generated for each seed.

Step 3: Evaluation and Fitness—The fitness of each branch is assessed by evaluating its performance concerning the optimization problem at hand. The fitness function measures how well a particular branch (representing a potential solution) satisfies the objectives of the problem. In mathematical terms, the fitness function can be represented as 𝒇(*B_ij_*), where 𝒇 is the fitness function and *B_ij_* is a branch.

Step 4: Completion and Pruning; Competition and Pruning—Not all branches survive; there is competition among them. DGO retains the best-performing branches based on their fitness scores while pruning less successful branches. This competition and pruning process ensures that only the most promising solutions continue to the next iteration. Mathematically, the competition and pruning process can be expressed as follows: *SurvivedBranches* = *SelectBest(Branches,PruningThreshold*), where *SelectBest* is a function that retains the top-performing branches, and *PruningThreshold* is a parameter defining the survival threshold.

Step 5: Iteration and Evaluation—DGO repeats this process for multiple iterations, iteratively branching, evaluating, competing, and pruning. The iterations enable the algorithm to converge towards optimal solutions gradually.
*S*^(*t*+1)^ = *SelectBest*(*B*(*S^t^*), *PruningThreshold*

Here, *S*^(*t*)^ represents the set of seeds at iteration *t*, and *S*^(*t*+1)^ represents the updated set of seeds after iteration *t* + 1.

Step 6: Termination Criteria—The algorithm terminates when predefined stopping criteria are met, such as a maximum number of iterations or convergence to a satisfactory solution. At this point, the best solution(s) among the surviving seeds is returned as the algorithm’s output.

Thus, DGO brings several advantages to the table. DGO’s branching strategy allows it to explore diverse solution spaces efficiently, reducing the chances of getting stuck in local optima. DGO’s ability to retain and promote top-performing branches fosters the search for global optima in complex, non-convex landscapes. DGO’s collaborative approach helps it navigate noisy fitness landscapes, making it suitable for real-world optimization problems. DGO’s adaptability to various optimization challenges and domains positions it as a versatile optimization technique.

#### 2.2.1. Mathematical Representation of DGO in Action

This section discusses functions to understand the workings of DGO. This mathematical representation demonstrates how DGO operates iteratively to explore and optimize the solution space. A mathematical function ***f***(*x*) is used to illustrate DGO’s branching, evaluation, and pruning process.
Consider the function ***f***(*x*) = *x*^2^ − 4*x* + 2

Initialize a set of seeds representing potential solutions.
*S* = {2.0, 2.5, 3.0}

Generate branches around each seed by introducing slight perturbations. For instance, for *S* = 2.0, generate branches as follows:*B*(2.0) = {1.9, 2.0, 2.1}

Calculate the fitness of each branch using the function ***f***(*x*). The fitness function can be expressed as ***f***(*B_ij_*)*,* where *B_ij_* represents a branch.

For example, the fitness of a branch B = 1.9 can be calculated as follows:***f***(1.9) = (1.9)2 − 4(1.9) + 4 = 0.21

Select the best-performing branches based on their fitness scores, i.e., prune branches with lower fitness. For instance, the branch with the highest fitness (smallest ***f***(*x*) value) may be retained and the others may be pruned.

The surviving branches become the new seeds for the next iteration. Iteratively repeat the branching, evaluation, competition, and pruning steps.

The algorithm terminates when predefined stopping criteria are met. This could be a maximum number of iterations or reaching a satisfactory solution.

The best solution(s) among the surviving seeds is returned as the algorithm’s output.

Figure 1a represents the overall flow of the DGO algorithm. Figure 1b depicts the pseudocode based on the algorithm.

Figure 1a and the pseudocode may be described as follows:

Step 1: Initialization of Seeds—initialize_seeds() returns an initial population of solutions.

Step 2: Evaluate Fitness—evaluate_fitness(population) computes fitness values for each solution in the population.

Step 3: Random Perturbation—random_perturbation(solution) introduces small, random changes to a solution.

Step 4: Competition and Pruning—competition_and_pruning(population, fitness_values) selects solutions based on competition and pruning mechanisms.

Step 5: Knowledge Transfer—knowledge_transfer(selected_population) allows solutions to transfer knowledge to improve each other.

Step 6: Iteration and Evolution—iteration_and_evolution(current_population) performs one iteration of the DGO algorithm, including fitness evaluation, competition and pruning, and knowledge transfer.

Step 7: DGO Algorithm—DGO_algorithm(max_generations) is the main function that initializes the population, iteratively performs evolution, and checks for convergence. It returns the optimal solution if convergence is met within the specified number of generations.

Step 8: Convergence Check—convergence_condition_met(population) checks whether the convergence condition is met, indicating if further iterations are required.

Step 9: Optimal Solution Retrieval—get_optimal_solution(updated_population) retrieves the optimal solution when convergence is met.

Step 10: Best Solution Retrieval—get_best_solution(population) retrieves the best solution after the specified number of generations.

#### 2.2.2. DGO and the Local Minima Problem

DGO is designed to address the issue of local minima, which can hinder the search for the global minimum in complex optimization problems. DGO employs several strategies to navigate through the search space and mitigate the impact of local minima:DGO introduces genetic diversity by incorporating mutation operations. Mutation involves making small, random changes to candidate solutions. This helps prevent premature convergence to local minima, as individuals in the population explore nearby regions of the search space.DGO balances exploration and exploitation. While it encourages the exploration of new regions through mutation and crossover, it also exploits promising solutions by selecting them to be part of the next generation. This allows DGO to benefit from both exploration (escaping local minima) and exploitation (refining promising solutions).DGO uses crossover operations to exchange information between candidate solutions. Combining traits from multiple individuals creates offspring that inherit characteristics from their parents. This sharing of genetic information allows DGO to escape local minima by combining the strengths of different solutions.Many variants of DGO incorporate adaptive strategies. These mechanisms allow DGO to adjust its behavior during optimization dynamically. Suppose the algorithm detects that it is converging too quickly or getting trapped in local minima. In that case, it can shift its focus toward exploration by increasing mutation rates or altering other parameters.DGO preserves good solutions throughout the optimization process. Even if a local minimum is encountered, DGO does not immediately discard it. Instead, it retains the solution as part of the population, ensuring that valuable information is not lost. This helps prevent the complete abandonment of promising solutions.Some DGO variants incorporate memory mechanisms to store historical information about solutions and their performance. This memory can guide the optimization process and prevent revisiting regions of the search space that have been explored extensively.DGO may dynamically adjust its parameters, such as mutation rates and selection pressure, to adapt to the optimization problem’s characteristics. This adaptability helps the algorithm effectively navigate challenging landscapes with local minima.

Figure 2 depicts how DGO solves the local minima problem.
Initialization—DGO begins by initializing a population of candidate solutions.Evaluate Fitness—The fitness of each candidate solution is evaluated based on the objective function.Select Individuals for Reproduction—DGO selects individuals from the population, applies crossover and mutation operations, and generates offspring. Perturbation refers to diverse strategies to introduce disturbances either to the entire population or specific solution subsets, whereas mutation is a specialized genetic operator employed during reproduction, targeting individual solutions. Perturbation is concerned with the ability to sustain diversity and forestall premature convergence, while mutation is geared towards fostering genetic diversity in offspring to explore uncharted regions of the search space.Fitness Evaluation for Offspring—The fitness of the offspring is evaluated.Population Replacement—A portion of the current population is replaced with the offspring.Adaptive Strategies/Global Exploration Check—DGO may employ adaptive mechanisms to dynamically adjust its behavior.Termination Criteria—The optimization process continues until termination criteria are met, which could include a maximum number of iterations, convergence threshold, or other conditions.End—The algorithm concludes when it either finds a solution that meets the termination criteria or reaches a predefined stopping point.

**Figure 2 biomimetics-09-00130-f002:**
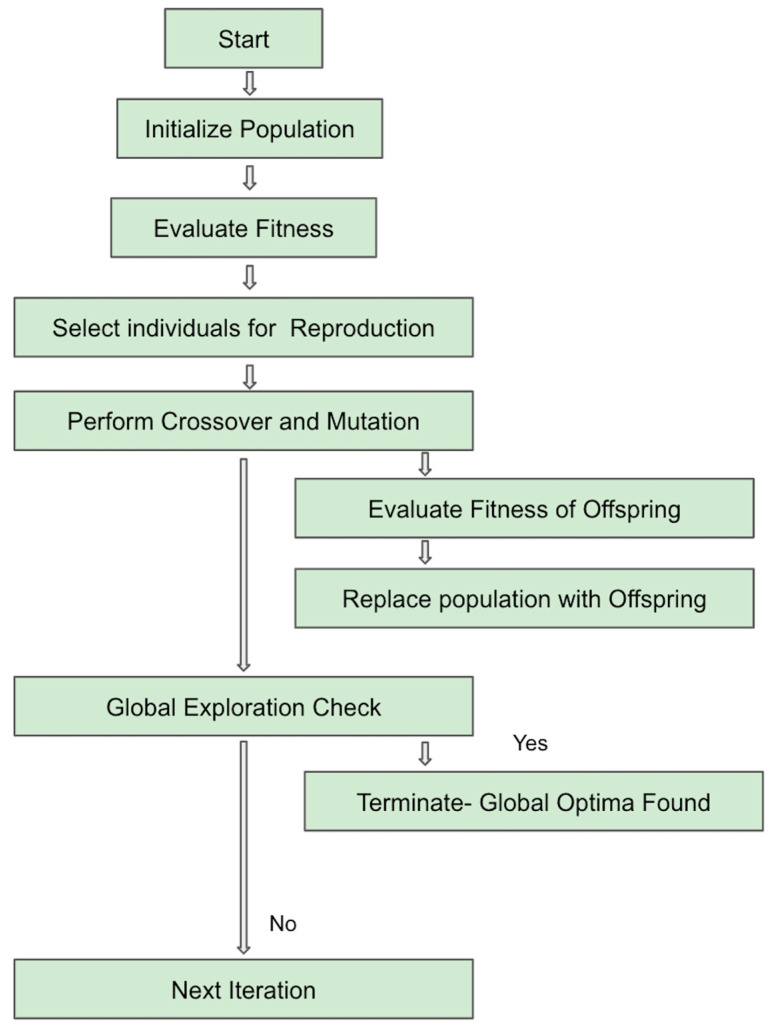
Flowchart depicting how DGO attains global minima.

In Dendritic Growth Optimization (DGO), the crossover-like operation promotes collaboration among solutions, facilitating the exchange of beneficial traits and knowledge transfer. This mechanism enhances the overall quality of solutions by allowing promising characteristics to influence neighboring solutions, contributing to convergence in high-performing solution regions. Serving as a form of collaborative exploration, crossover in DGO prevents premature convergence by maintaining diversity within the population. Inspired by dendritic branching patterns, this operation emulates natural information exchange, guiding the collective adaptation of solutions toward optimal configurations. While DGO’s crossover-like process parallels genetic algorithms, its focus on collaborative exploration aligns with the algorithm’s dendritic inspiration, fostering synergy among solutions in navigating the solution space.

Handling many decision variables in optimization problems can increase complexity, and finding the optimal solution efficiently becomes crucial. However, several strategies can be adopted to solve this challenge. Adaptive population sizing can be implemented instead of keeping the population size constant. This means dynamically adjusting the population size based on the characteristics of the optimization problem. For instance, the population size may be increased when dealing with many decision variables to enhance exploration. Moreover, the parameters can be fine-tuned to suit the specific characteristics of the optimization problem. Parameters such as mutation, crossover, and growth rates can be adjusted to improve convergence speed and exploration efficiency. Parallel computing can be particularly beneficial when dealing with many decision variables, as parallelization allows for simultaneous exploration of different regions of the solution space. The optimization problem can also be decomposed into smaller sub-problems. Each sub-problem can be optimized separately and then combined. This approach can be effective when dealing with high-dimensional spaces, as it reduces the overall complexity. The algorithm can dynamically adjust crossover operators based on the optimization’s progress, emphasizing operators that show effectiveness in high-dimensional spaces.

The novelty of DGO can be further expressed using the following attributes:Regarding the Branching Mechanism, while it is true that branching involves generating multiple variants (branches) from a seed, the key distinction lies in how DGO achieves branching. In DGO, the branching mechanism is inspired by natural dendritic growth, emphasizing a more organic and nature-inspired approach.The concept of Dendritic Growth itself, as DGO introduces a unique concept by mimicking dendritic growth patterns found in nature. Dendritic growth involves the iterative development of branching structures, and DGO adapts this concept to explore the solution space. This organic, hierarchical growth is distinct from traditional crossover/mutation operations.DGO involves a competition and pruning mechanism, where solutions compete for survival based on their fitness. The pruning process helps refine the population, focusing on promising branches. This combination of branching, competition, and pruning differentiates DGO from other evolutionary algorithms.Moreover, DGO draws inspiration from the branching and growth observed in biological systems, emphasizing the adaptation of natural processes for optimization. This biological inspiration differentiates it from algorithms that primarily focus on mathematical operations like crossover and mutation.

While increasing diversity is a common strategy in optimization algorithms, the novelty in DGO lies in increasing diversity and in the specific mechanism used, mimicking dendritic growth. The emphasis is on creating a population structure that organically adapts to the problem space. DGO introduces complexity inspired by natural systems, bringing sophistication beyond simple crossover and mutation. The interplay between branching, competition, and pruning adds a layer of intricacy to the optimization process. Also, DGO can be seen as a holistic optimization strategy that integrates multiple biological-inspired concepts into a coherent framework. This includes branching, competition, and pruning, creating a synergy beyond traditional algorithms. In summary, while the branching concept might share similarities with certain aspects of other algorithms, the holistic integration of dendritic growth, competition, and pruning, inspired by natural systems, contributes to the novelty of DGO. The goal is to provide a more biologically inspired, adaptive, and complex optimization approach.

## 3. Results and Observations

In this section, the study discuss the datasets that have been considered for the study followed by the performance evaluation metrics to assess the performance of the models before and after the optimization algorithm was implemented. The study also highlights a comparative analysis of the proposed work with some of the existing works. An observation section has been included to discuss the key findings.

### 3.1. Datasets

For analyzing the strength of the proposed optimization technique, two popular datasets have been considered. The Diabetes dataset is a prominent benchmark dataset in machine learning, particularly in healthcare and medical research. It comprises 442 data points, each representing a patient. The data are derived from a study conducted by Bradley Efron, Trevor Hastie, Iain Johnstone, and Robert Tibshirani, aiming to explore the relationship between patient attributes and diabetes progression [32]. For modeling purposes, the dataset includes ten numerical features, such as age, sex, body mass index (BMI), and blood serum measurements like insulin levels and triglycerides. The target variable in this dataset is a quantitative measure of diabetes progression, specifically the numeric disease progression score one year after baseline data collection. The Breast Cancer dataset, also known as the “Breast Cancer Wisconsin (Diagnostic)” dataset or “BCW dataset”, is widely used for binary classification tasks, particularly in breast cancer diagnosis. It encompasses 569 data points, each representing a breast biopsy sample collected by Dr. William H. Wolberg and colleagues at the University of Wisconsin Hospitals, Madison [33]. This dataset originates from medical research aimed at enhancing breast cancer diagnosis and treatment. For each data point, the dataset includes 30 numerical features derived from cell nuclei characteristics observed in microscopic images of breast tumor biopsy samples. These features encompass parameters such as mean radius, texture, and smoothness. The target variable in the Breast Cancer dataset is binary, indicating tumor diagnosis—0 for benign (non-cancerous) and 1 for malignant (cancerous). For performing the analysis, the data are split into 80% training set and 20% test set.

### 3.2. Performance Evaluation Metrics

Performance evaluation metrics are vital tools in assessing the effectiveness of machine learning models. They provide insights into how well a model performs and its ability to make accurate predictions. Here, the study explores five key performance metrics: Accuracy, Precision, Recall, F1 Score, and ROC Curves.
Accuracy: Accuracy is one of the most straightforward metrics, measuring the proportion of correctly predicted instances out of the total predictions. It is suitable for balanced datasets but can be misleading when dealing with imbalanced datasets.
Accuracy = (True Positives + True Negatives)/Total PredictionsPrecision: Precision assesses the model’s ability to correctly identify positive instances. It measures the ratio of true positive predictions to the total predicted positives. High precision indicates fewer false positives.
Precision = True Positives/(True Positives + False Positives)Recall (Sensitivity or True Positive Rate): Recall evaluates the model’s capability to identify all relevant instances in the dataset. It calculates the ratio of true positives to the total actual positives. High recall implies fewer false negatives.
Recall = True Positives/(True Positives + False Negatives)F1 Score: The F1 Score combines precision and recall into a single metric, balancing the trade-off between false positives and false negatives. It is particularly useful when dealing with imbalanced datasets.
F1 Score = 2 × (Precision − Recall)/(Precision + Recall)ROC Curves (Receiver Operating Characteristic Curves): ROC curves are valuable for binary classification models. They illustrate the trade-off between true positive rate (sensitivity) and false positive rate (1-specificity) at various thresholds. The area under the ROC curve (AUC-ROC) quantifies the model’s overall performance, with a higher AUC indicating better performance. AUC-ROC measures the area under the ROC curve, ranging from 0 to 1, where 0.5 represents random guessing and 1 signifies perfect classification.

### 3.3. Results

For evaluating the performance of the proposed DGO, two datasets have been considered, i.e., Diabetes data and Breast Cancer data. The study deploys several machine learning and deep learning algorithms, such as K-Nearest Neighbor (KNN), Logistic Regression (LR), Artificial Neural Networks (ANN), Support Vector Machines (SVM), Random Forests (RF), Convolutional Neural Networks (CNN), and CNN combined with Long–Short Term Memory (CNN-LSTM). Moreover, the study also incorporates results from metaheuristic algorithms like Particle Swarm Optimization (PSO), Non-dominated Sorting Genetic Algorithm-II (NSGA-II), Ant Colony Optimization (ACO), and Genetic Algorithm (GA) for comparing the performance of DGO. After deploying the algorithms on the designated dataset, they are integrated with the optimization technique to assess whether there is an enhancement in performance. Table 1 and Table 2 present performance evaluation of the algorithms before and after DGO was applied on the Diabetes dataset.

From Table 1, it can be observed that 1D CNN exhibits the best performance in terms of accuracy followed by CNN-LSTM and ANN. KNN shows the least impressive performance among the suite of algorithms considered for the study.

**Table 2 biomimetics-09-00130-t002:** Performance Evaluation of ML models after applying DGO.

Model	Accuracy	Precision	Recall	F1-Score	Time to Run (s)
KNN + DGO	0.71	0.70	0.70	0.69	1.32
LR + DGO	0.78	0.75	0.75	0.75	1.65
ANN + DGO	0.81	0.79	0.78	0.79	5.73
SVM + DGO	0.78	0.76	0.77	0.78	8.54
RF + DGO	0.75	0.76	0.70	0.70	3.18
CNN + DGO	0.81	0.82	0.81	0.80	9.23
CNN + LSTM + DGO	0.78	0.79	0.77	0.80	20.09
PSO + DGO	0.83	0.81	0.78	0.81	26.34
NSGA-II + DGO	0.79	0.81	0.81	0.79	41.04
ACO + DGO	0.83	0.82	0.80	0.82	35.55
GA + DGO	0.81	0.82	0.80	0.79	29.26

From Table 2, it is clearly observed that once DGO is applied to the suite of ML algorithms, the performance improvises significantly. The accuracy of CNN increases from 0.79 to 0.81, the accuracy of CNN-LSTM increases from 0.76 to 0.78, and the accuracy of ANN increases from 0.75 to 0.81. This validates that the proposed algorithm can be successfully deployed for optimization purpose. The time to run the algorithms increases significantly with the deployment of DGO. We observe that NSGA-II combined with DGO takes the maximum time to run following ACO + DGO and GA + DGO. KNN + DGO takes the least amount of time to run.

Figure 3 depicts ROC plots for some of the algorithms considered for the analysis with the DGO algorithm. It is observed that the plots with DGO are closer to the top left of the graph and have a higher score. This depicts that the algorithms combined with DGO show better performance.

Table 3 and Table 4 present performance evaluation of the algorithms before and after DGO was applied to the Breast Cancer dataset.

From Table 3, it can be observed that LR and ANN exhibit the best performance in terms of accuracy followed by CNN-LSTM, CNN, and SVM. KNN ad RF show the least impressive performance among the suite of algorithms considered for the study for the breast cancer dataset.

**Table 4 biomimetics-09-00130-t004:** Performance Evaluation of ML models after applying DGO.

Model	Accuracy	Precision	Recall	F1-Score	Time to Run (s)
KNN + DGO	0.94	0.93	0.98	0.95	1.87
LR + DGO	0.98	0.92	0.99	0.97	2.03
ANN + DGO	0.97	0.97	0.98	0.98	5.37
SVM + DGO	0.95	0.97	0.96	0.96	9.12
RF + DGO	0.96	0.95	0.98	0.96	2.89
CNN + DGO	0.96	0.97	0.96	0.97	13.52
CNN + LSTM + DGO	0.97	0.97	0.98	0.98	29.11
PSO + DGO	0.96	0.94	0.95	0.94	38.06
NSGA-II + DGO	0.96	0.98	0.92	0.92	48.66
ACO + DGO	0.97	0.94	0.95	0.93	40.98
GA + DGO	0.98	0.97	0.98	0.95	37.83

From Table 4, it is clearly observed that once DGO is applied to the suite of ML algorithms as well, the performance improvises significantly. The accuracy of LR increases from 0.97 to 0.98, while ANN stays the same. The performances for CNN and CNN-LSTM increase from 0.94 and 0.96 to 0.96 and 0.97. For most of the algorithms, there seems to be an increase in the performance accuracy. This validates that the proposed algorithm can be successfully deployed for optimization purpose. The time to run the algorithms increases significantly with the deployment of DGO. We observe that NSGA-II combined with DGO takes the maximum time to run following ACO + DGO and GA + DGO. KNN + DGO takes the least amount of time to run.

Figure 4 depicts ROC plots for some of the algorithms considered for the analysis with the DGO algorithm. It is observed that the plots with DGO are closer to the top left of the graph and have a higher score. This depicts that the algorithms when combined with DGO yield better performance.

### 3.4. Comparative Analysis

In this section, the comparative analysis of the proposed work along with some previous related works is reported. Table 5 summarizes the overall comparison.

### 3.5. Observations

Based on the methodology and experimental analysis, the following observations can be made.
DGO is a newly proposed optimization algorithm inspired by natural branching patterns, making it unique in its approach to solving complex optimization problems. Its innovative architecture and principles set it apart from traditional optimization techniques.DGO’s architecture is designed to be adaptable and flexible, allowing it to be applied to a wide variety of optimization challenges and tasks. This flexibility makes DGO a promising candidate for diverse applications across different domains.DGO addresses the local minima problem by employing a combination of strategies that promote diversity, balance exploration and exploitation, encourage information exchange between solutions, and adapt to the optimization problem’s dynamics. These mechanisms work together to help DGO escape local minima and converge toward the global minimum in complex optimization scenarios.When DGO is applied to various machine learning and deep learning algorithms, it consistently leads to performance improvements. Metrics such as precision, accuracy, recall, F-1 score, and ROC curve all show improved performances.DGO’s effectiveness is validated through extensive testing on two distinct datasets, i.e., the Diabetes dataset and the Breast Cancer dataset. Both datasets serve as benchmarks to assess DGO’s performance on different data types and domains.DGO’s robustness is evident in its ability to tackle various optimization challenges effectively. It can adapt to complex problem scenarios and deliver meaningful results.DGO’s scalability is a key attribute that suggests its potential for application in large-scale and intricate problem domains. It can handle complex optimization tasks that involve a significant volume of data and parameters.

Based on the comparative analysis (Table 5), it is observed that DGO is a novel and robust optimization technique, which addresses several limitations of already existing optimization algorithms. It is well suited to address complex problems, and exhibits superiority over existing algorithms in optimization field in the following manners:DGO excels in efficiently exploring complex solution spaces. It leverages its inspiration from natural dendritic growth patterns to adapt and navigate diverse optimization landscapes effectively. This adaptability sets it apart from some other nature-inspired methods, which might struggle in highly nonlinear or multi-modal spaces.DGO often demonstrates faster convergence rates compared to certain traditional optimization algorithms and other nature-inspired techniques. Its ability to rapidly identify promising regions in the solution space contributes to quicker optimization, saving computational time and resources.DGO showcases versatility by consistently delivering strong performance across various problem domains and datasets. This adaptability extends its utility beyond specific niches and allows it to address a wide array of optimization challenges, whereas some other techniques may require fine-tuning for different domains.DGO’s innovative nature-inspired approach, drawing inspiration from dendritic growth patterns, introduces a fresh perspective to optimization. This uniqueness can be advantageous for solving novel or unconventional optimization problems where traditional methods might not be directly applicable.DGO strikes a balance between global and local optimization. It is capable of efficiently searching for global optima while also navigating local optima effectively. This balance is crucial for solving complex optimization tasks with multiple extrema, which can be challenging for some other algorithms.DGO exhibits robustness in handling noisy data and complex problem scenarios. Moreover, its scalability makes it suitable for both small and large-scale optimization problems, providing consistent performance across a wide spectrum of applications.DGO’s adaptability and efficiency extend its utility to various fields, including machine learning, logistics, engineering, and beyond. This interdisciplinary applicability broadens its potential impact compared to optimization techniques tailored to specific domains.

## 4. Limitations and Constraints

Dendritic Growth Optimization (DGO) is a promising optimization algorithm, but it also faces several challenges and limitations DGO can be computationally intensive, especially for complex optimization problems with a large number of variables and constraints. This can lead to long optimization times and increased resource requirements, limiting its practicality for certain applications. Like many optimization algorithms, DGO’s performance is sensitive to the choice of hyperparameters, such as mutation rates and convergence criteria. Finding the right set of hyperparameters can be challenging and may require manual tuning. DGO, like most optimization techniques, is not guaranteed to find the global optimum in all cases. It can still be susceptible to getting trapped in local optima, which are suboptimal solutions within a limited region of the search space. DGO is a relatively new optimization approach, and its theoretical foundations may not be as well-established as some traditional optimization methods. This can make it difficult to provide formal guarantees of convergence or optimality. While DGO has shown promise in various applications, it may not have been extensively tested in all possible real-world scenarios. Its performance across diverse problem domains is an area of ongoing research. DGO may not perform optimally in optimization problems with noisy or stochastic objective functions, where the fitness landscape can change unpredictably. It might require additional adaptations to handle such scenarios effectively. Handling extremely large-scale optimization problems can still be a challenge, and there may be limitations in terms of memory and computational resources. While DGO is scalable to some extent, there are practical limitations in terms of handling extremely high-dimensional optimization problems. Implementing DGO can be more complex compared to using off-the-shelf optimization libraries for well-known algorithms. It may require expertise in both the optimization domain and the specific problem being addressed. Despite these challenges, Dendritic Growth Optimization continues to evolve and gain recognition as a valuable tool in the optimization field. Researchers are actively working to address these limitations and refine the algorithm for improved performance and broader applicability.

## 5. Conclusions and Future Work

In this paper, the novel Dendritic Growth Optimization (DGO) algorithm, inspired by natural branching patterns, has been introduced to tackle complex optimization problems across various domains. DGO’s unique architecture and working principles provide an effective approach to overcoming local minima, a common challenge in optimization. Extensive experiments were conducted using diverse datasets, including Breast Cancer and Diabetes datasets. DGO was evaluated alongside various machine learning algorithms, showcasing its consistent performance improvements in terms of accuracy, precision, recall, F1 score, and ROC curve. These results demonstrate DGO’s potential for enhancing machine learning and data analysis tasks. Comparative analyses with existing optimization methods highlighted DGO’s advantages, particularly its ability to generalize across datasets and consistently enhance results. This positions DGO as a valuable tool for researchers and practitioners seeking improved optimization solutions. DGO’s applicability extends beyond machine learning, with potential applications in agriculture, environmental conservation, game AI, robotics, and more. Its adaptability and robustness make it a promising choice for optimizing complex systems and processes in various real-world scenarios. While DGO presents exciting possibilities, it faces challenges like parameter tuning and scalability, which require further refinement.

In the future, it might be interesting to investigate the potential of hybridizing DGO with other optimization algorithms or machine learning techniques. Combining DGO’s unique strengths with those of other methods can lead to even more robust and versatile optimization solutions. It may also be necessary to continuously benchmark DGO against state-of-the-art optimization algorithms and update performance comparisons to ensure its competitiveness and relevance in the evolving landscape of optimization techniques. DGO’s effectiveness can be demonstrated by solving complex real-world problems and providing practical insights.

## Figures and Tables

**Figure 1 biomimetics-09-00130-f001:**
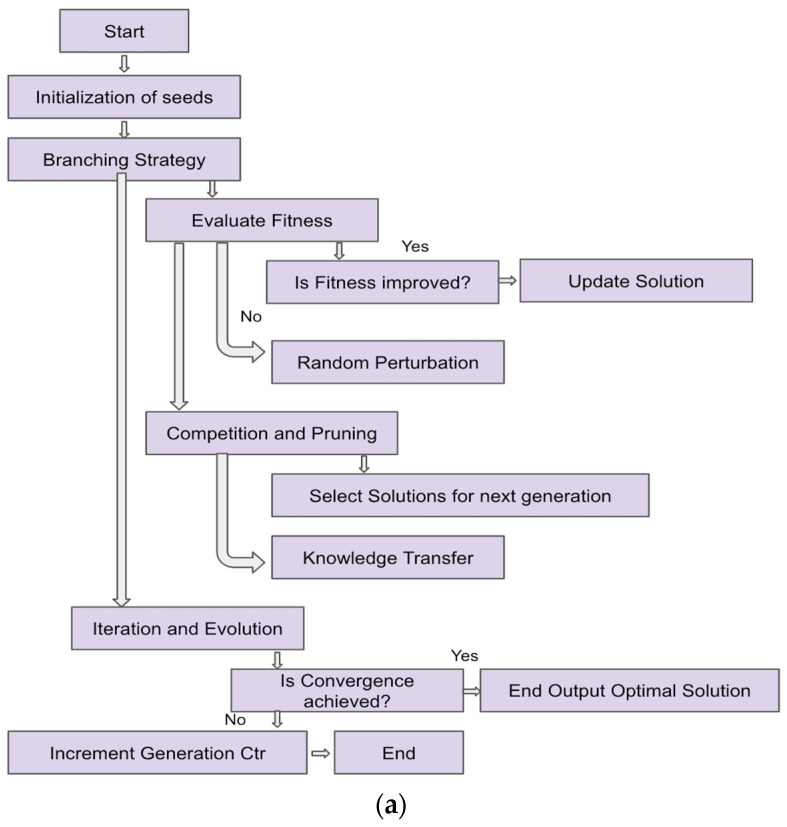
(**a**) Flowchart for the proposed DGO algorithm; (**b**) pseudocode for the proposed DGO algorithm.

**Figure 3 biomimetics-09-00130-f003:**
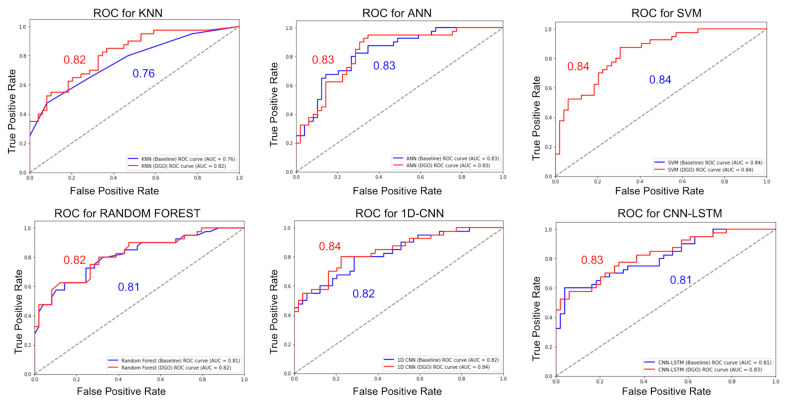
ROC curves comparison with DGO algorithm (Diabetes).

**Figure 4 biomimetics-09-00130-f004:**
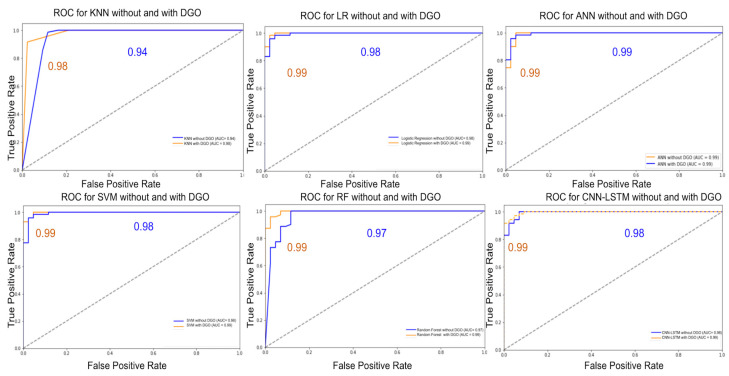
ROC curves comparison with DGO algorithm (Breast Cancer).

**Table 1 biomimetics-09-00130-t001:** Performance Evaluation of ML models before applying DGO.

Model	Accuracy	Precision	Recall	F1-Score	Time to Run (s)
KNN	0.69	0.68	0.69	0.68	0.23
LR	0.73	0.69	0.72	0.71	0.31
ANN	0.75	0.71	0.75	0.73	2.34
SVM	0.74	0.69	0.78	0.73	4.12
RF	0.72	0.69	0.68	0.68	0.79
CNN-1D	0.79	0.78	0.78	0.77	6.44
CNN-LSTM	0.76	0.77	0.76	0.78	14.27
PSO	0.81	0.80	0.77	0.77	18. 14
NSGA-II	0.76	0.78	0.79	0.78	26.14.
ACO	0.81	0.81	0.81	0.79	21.45
GA	0.79	0.80	0.79	0.80	20.87

**Table 3 biomimetics-09-00130-t003:** Performance Evaluation of ML models before applying DGO.

Model	Accuracy	Precision	Recall	F1-Score	Time to Run (s)
KNN	0.92	0.93	0.95	0.94	0.13
LR	0.97	0.97	0.98	0.97	0.19
ANN	0.97	0.97	0.98	0.97	2.02
SVM	0.94	0.92	0.95	0.96	5.03
RF	0.93	0.93	0.97	0.95	0.45
CNN-1D	0.94	0.96	0.96	0.95	8.18
CNN-LSTM	0.96	0.97	0.97	0.97	19.37
PSO	0.95	0.93	0.92	0.94	25.21
NSGA-II	0.94	0.96	0.93	0.93	34.08
ACO	0.96	0.95	0.92	0.95	29.81
GA	0.96	0.92	0.94	0.94	26.03

**Table 5 biomimetics-09-00130-t005:** Comparative analysis of proposed work with some related works.

Author and Year	Proposed Work	Methodology/Parameters	Results
Seyyedabbasi and Kiani 2023 [25]	Nature-inspired optimization algorithm, Sand Cat Swarm Optimization (SCSO),	More then 20 test functions of CEC2019 benchmark functions	SCSO performed best in 63.3% of the test functions
Yuan et al., 2023 [27]	Bionic optimization algorithm, Coronavirus Mask Protection Algorithm (CMPA),	CEC2020 suite problems, state-of-the-art metaheuristic algorithms	Mass and deflection improved by 16.44% and 7.49%
Shandilya et al., 2023 [34]	Modified Firefly Optimization Algorithm-Based IDS	Early detection of suspicious nodes, event management schemes	Suspicious nodes reduced by 60–80%
Singh et al., 2023 [35]	Nature-inspired computing for detecting glaucoma in retinal fundus images	Particle Swarm Optimization (PSO), Artificial Bee Colony (ABC), and Binary Cuckoo Search (BCS)	BCS shows up to 98.46% accuracy
Yuan et al., 2022 [36]	Alpine skiing optimization	Mathematical modelling, performance evaluation	Braking efficiency factor is improved by 28.446%
Husnain and Anwar 2022 [37]	Intelligent Probabilistic Whale Optimization Algorithm	Clustering in vehicular ad hoc networks	75% improvement in cluster optimization
Patil et al., 2023 [38]	Predicting type-2 diabetes using optimization	Stacking-based non-dominated sorting genetic algorithm (NSGA-II)	Accuracy is 81% on Pima dataset and 89% on collected data
Proposed Work	Dendritic Growth Optimization (DGO) Algorithm	KNN, LR, ANN, SVM, RF, CNN, CNN-LSTM, PSO, NSGA-II, ACO, GA	Significant improvement in performance (up to 83% accuracy in dataset1, and 98% accuracy in dataset2)

## Data Availability

Information on the datasets is included in Section 3.

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
