# Peer review of "Dendritic Growth Optimization: A Novel Nature-Inspired Algorithm for Real-World Optimization Problems"

_biomimetics, 2024, doi:10.3390/biomimetics9030130_

Round 1

Reviewer 1 Report

Comments and Suggestions for Authors

This paper proposed a novel algorithm inspired by natural branching patterns for real-world optimization problems. Overall, this paper is well written and has solid contribution. However, several issues should still be addressed to further improve the quality of this paper. Below are some comments for the author to consider:

1. In section 3, the author presented two case studies to demonstrate the application of the proposed DGO algorithm. However, no data source of the two datasets (the Diabetes and Breast Cancer datasets) was provided. In my opinion, the author should also provide some literature references or put the datasets in the supplementary material. In this way, the readers can better understand the results of the applications of the proposed algorithm.

2. In line 331, the rest of the sentence "The figure also includes a" was missing. Please improve that.

3. The text in Figure 3 and 4 was too small to read. Please improve the quality of these two figures.

4. The literature review part could be further extended. Currently, the bio-inspired topology optimization algorithm is also frequently used to achieve design optimization in different applications. From this point of view, the author is also recommended to mention the topology optimization algorithm in the Introduction section as an important aspect of bio-inspired optimization algorithms. Below are some recommended related work of bio-inspired topology optimization algorithms:

“Design of topology optimized compliant legs for bio-inspired quadruped robots”. https://doi.org/10.1038/s41598-023-32106-5

"Topology optimization of plate/shell structures with respect to eigenfrequencies using a biologically inspired algorithm". https://doi.org/10.1080/0305215X.2018.1552952

Author Response

Dear Prof.,

Thank you for your useful comments and suggestions on the modification of the manuscript. The manuscript has been modified accordingly, and detailed corrections are listed in the attachment point by point:

Reviewer 2 Report

Comments and Suggestions for Authors

This research proposes a new algorithm called DGO which was inspired by the branching and connection mechanisms observed in dendritic structures. Experimental results suggest that the selected ML models can be improved by integrating with DGO over two datasets. While DGO sounds unique, there are some major issues.

1. Literature review should examine most of the advanced optimisation algorithms and clearly highlight the knowledge gap which helps to inspire the proposed method. However, take section 2.1 as example, although it reviews a number of existing algorithms, it must be better structured to specify the gap, i.e. what is missing, by evaluating instead of summarising existing approaches.

2. Although DGO is claimed as novel, its key mechanism is not significantly different from some of the existing optimisation algorithm. For example, step 2 branching strategy, multiple branches are developed from the seed. Without specifying how branching happens quantitatively, I cannot tell if it is different from any initial randomisation. Moreover, step 4 completion and pruning. Again it does not sound new as it is merely keeping the best solution and keep it to the next generation which is commonly seen in genetic algorithm.

3. The mechanisms of DGO are described as novel however there is no mathematical formula/equation to tell how new is it. To prove that a method is new, each step of the proposed algorithm must be clearly presented and explained as well as compared against its closet counterpart.

4. Section 3.3 reports the comparison results. It can be seen that, integrating with DGO, the performance of most of the selected ML models can be improved. However, it is not clear the statistical significance of such improvement. Also, how the comparisons were done? how many replications? Were these ML modes fine-tuned? Was DGO fine-tuned as well? What are the key parameters of DGO?

5.  Section 3.4 presents comparative analysis. But not clear how such comparison was done. Table 5 does not show all the necessary information but remains very descriptive. More supporting evidence must be reported.

6. Given a lack of evidence to prove the novelty of the proposed algorithm, Section 4 does not sound convincing. This section could be removed to give room for more clarifications over the development and parameter setting of the proposed DGO. More datasets could be used to support the comparison.

Comments on the Quality of English Language

The presentation is fair.

Author Response

(The authors gave the same response as above.)

Reviewer 3 Report

Comments and Suggestions for Authors

1. Regarding the principle of dendritic cells, a schematic diagram is suggested to be given to illustrate the optimization mechanism, the description with only words is not easy to understand;

2. The pseudo-code in Figure 1 should be improved referring to the articles of IEEE journals or Evolutionary and Computation(MIT), and the flowchart and the figures in conclusion should also be improved;

3. The direction of change will be increased obviously when the dendritic cell approach encounters a large number of decision variables. How can seek the optimal solution as quickly as possible even the population size is kept constant? The author should explain that.

4. The DGO is a perturbation, the difference between this perturbation and the mutation in Fig. 2 should be explained in detail?

5. The Crossover also plays a crucial role in the algorithm, the authors should add more description of the Crossover to help readers better understand the mechanism of DGO.

6. Besides the comparison results, it is suggested to compare DGO with PSO, NSGA, and other algorithms using the standard test functions.

Author Response

(The authors gave the same response as above.)

Round 2

Reviewer 1 Report

Comments and Suggestions for Authors

The authors have revised the manuscript according to my comments. Therefore, I recommend to publish this paper in this journal.

Author Response

Thank you very much for the comment. 

Reviewer 2 Report

Comments and Suggestions for Authors

Thanks for your effort in revising the paper. In general, more clarifications have been provided in the revised manuscript and this certainly helps to improve the value of this research. In particular, more explanations/details have been given to the working mechanism of DGO. While some of my concerns have been satisfactorily addressed, there are some issues as specified below.

1. The novelty of DGO lies in the adoption of new strategies which are different from other evolutionary algorithms. For example, the branching enables the generation of multiple variants (branches) of a certain solution (seed). However, such branching is done through the use of crossover/mutation operations to generate more new seeds from the original seed. There is no significant difference from, e.g. Genetic Algorithm and its variants which introduce more crossovered/mutated offsprings in each generation. The same can be done to other algorithms by increasing the diversity. In other words, the branching may sound new but its actual working mechanism is not. More clarifications are expected.

2. In relation to my first point above, by introducing more diversity into the solution exploration process, it surely can improve the optimisation outcome. However, this may add more burden to the computational effort. Hence, this research, in my opinion, will benefit from comparing optimisation effort (such as computing time) among DGO and other benchmarking methods.

Author Response

Thank you very much for the comment. Please find the response letter attached. 
